# Inhibition of the deubiquitinase USP8 corrects a Drosophila PINK1 model of mitochondria dysfunction

Sophia von Stockum[1], Alvaro Sanchez-Martinez[2], Samantha Corrà[3,4], Joy Chakraborty[3], Elena Marchesan[1], Lisa Locatello[3], Caterina Da Rè[3,4], Paola Cusumano[3,4], Federico Caicci[3], Vanni Ferrari[3], Rodolfo Costa[3,4], Luigi Bubacco[3], Maria Berica Rasotto[3], Ildiko Szabo[3], Alexander J Whitworth[2], Luca Scorrano[3,5], Elena Ziviani[1,3]

**Aberrant mitochondrial dynamics disrupts mitochondrial function and contributes to disease conditions. A targeted RNA interference screen for deubiquitinating enzymes (DUBs) affecting protein levels of multifunctional mitochondrial fusion protein Mitofusin (MFN) identified USP8 prominently influencing MFN levels. Genetic and pharmacological inhibition of USP8 normalized the elevated MFN protein levels observed in PINK1 and Parkin-deficient models. This correlated with improved mitochondrial function, locomotor performance and life span, and prevented dopaminergic neurons loss in *Drosophila* PINK1 KO flies. We identified a novel target antagonizing pathologically elevated MFN levels, mitochondrial dysfunction, and dopaminergic neuron loss of a *Drosophila* model of mitochondrial dysfunction.**

## Introduction

Mitochondria dysfunction plays critical role in neurodegenerative conditions affecting the elderly, such as Parkinson's disease (PD) (Moore et al, 2005; Bueler, 2009; Vives-Bauza et al, 2010a; Ryan et al, 2015). Mitochondria function directly correlates with mitochondria dynamics and balanced remodeling of the mitochondrial network through fission and fusion events to control mitochondria shape and ultrastucture. Intuitively, fusion maintains the mitochondrial network and allows intermixing of matrix contents, such as mtDNA and metabolites; fission is needed to populate new cells with new mitochondria (Twig et al, 2008b; Gomes & Scorrano, 2008; Malena et al, 2009) and plays a substantial role in the mitochondria quality control. A key aspect of mitochondrial quality control is a well-characterized process called mitophagy that segregates and selectively eliminates damaged mitochondria via autophagy (Twig et al, 2008a; Twig & Shirihai, 2011). During stress-induced mitophagy, the cytoplasmic protein Parkin, mutated in familial PD and encoding an E3 ubiquitin ligase (Shimura et al, 2000), translocates in a PINK1-dependent manner to dysfunctional mitochondria (Narendra et al, 2008; Vives-Bauza et al, 2010b; Ziviani et al, 2010). In this process, kinase PINK1, also mutated in familial PD (Silvestri et al, 2005), phosphorylates Parkin (Sha et al, 2010), its targets (Wang et al, 2011; Chen & Dorn, 2013), and ubiquitin itself (Koyano et al, 2014) promoting Parkin translocation (Narendra et al, 2010; Ziviani et al, 2010) and Parkin activity (Lazarou et al, 2013; Koyano et al, 2014; Zhang et al, 2014). On depolarized mitochondria, Parkin ubiquitinates the mitochondrial pro-fusion protein Mitofusin (MFN) (Gegg et al, 2010; Poole et al, 2010; Tanaka et al, 2010; Ziviani et al, 2010; Sarraf et al, 2013) leading to p97/VCP–mediated retrotranslocation and proteosomal degradation (Tanaka et al, 2010). In addition, Parkin ubiquitinates the mitochondrial protein translocase TOM20, mitochondrial VDAC/Porin and Fis1 (Sarraf et al, 2013), and it also promotes the degradation of Miro (Wang et al, 2011), a protein that couples mitochondria to microtubules. Selected mitochondria are, therefore, deprived of their pro-fusion protein MFN, isolating them from the mitochondrial network, before degradation via autophagy. This mechanism is consistent with observations showing that mitochondria cluster around the perinuclear area (Vives-Bauza et al, 2010b) and fragment before mitophagy (Twig et al, 2008a; Poole et al, 2008). Genetic studies in *Drosophila* showed that down-regulation of MFN or promotion of mitochondrial fission by expressing pro-fission protein DRP1 rescues Parkin KO phenotypes, and those of kinase PINK1 (Deng et al, 2008; Poole et al, 2008), which acts upstream of Parkin (Clark et al, 2006; Park et al, 2006; Yang et al, 2006). This genetic interaction can be in part explained biochemically by the fact that Parkin ubiquitinates MFN to control its steady-state levels (Gegg et al, 2010; Tanaka et al, 2010; Ziviani et al, 2010; Rakovic et al, 2011) that are elevated in Parkin and PINK1 KO models (Ziviani et al, 2010). Thus, interventions that restore MFN levels can ameliorate Parkin and PINK1 phenotypes, presumably by impinging on the numerous

[1]Fondazione Ospedale San Camillo, IRCCS, Venezia, Italy    [2]MRC Mitochondrial Biology Unit, Cambridge Biomedical Campus, Cambridge, UK    [3]Department of Biology, University of Padova, Padova, Italy    [4]Neurogenetics and Behavior of Drosophila Lab, Department of Biology, University of Padova, Padova, Italy    [5]Dulbecco-Telethon Institute, Venetian Institute of Molecular Medicine, Padova, Italy

Correspondence: elena.ziviani@unipd.it

MFN functions that in the fruit fly include both promotion of fusion and ER–mitochondria crosstalk (Debattisti et al, 2014).

To identify other mechanisms regulating MFN levels, we performed an RNA interference screen for deubiquitinating enzymes (DUBs) that affect steady-state levels of MFN. DUBs participate in important reversible signaling pathways (Salmena & Pandolfi, 2007) and are attractive druggable candidates (Hussain et al, 2009; Colland, 2010). We identified USP8, an evolutionary conserved DUB whose down-regulation correlates with decreased MFN levels. USP8 has previously been linked to PINK1/Parkin–dependent mitophagy in cell culture and under intoxicating conditions (Durcan et al, 2014), but no in vivo studies have been reported. Here, we demonstrate that in vivo under basal conditions, genetic and pharmacological inhibition of USP8 ameliorates *Drosophila* phenotypes deriving from loss of function of PINK1 and Parkin.

# Results

### A targeted siRNA screening identifies DUBs affecting MFN protein levels

Steady-state levels of MFN protein in *Drosophila* PINK1 or Parkin KO background are increased (Ziviani et al, 2010), and interventions that decrease MFN levels can ameliorate *Drosophila* PINK1 and Parkin phenotypes (Celardo et al, 2016; Deng et al, 2008; Poole et al, 2008). Given the importance of MFN in inter-organellar communication (Cosson et al, 2012; de Brito & Scorrano, 2008; Filadi et al, 2015) and mitophagy (Chen & Dorn, 2013), we set out to identify regulators of its steady-state levels. We designed an unbiased loss-of-function screen using dsRNA to inhibit the expression of 35 known or predicted fly DUBs. Fly DUBs were identified by domain similarity and based on the list of 79 human DUBs (Dupont et al, 2009) (Table 1). We transiently expressed flag-tagged MFN in S2R+ cells to mimic pathologically elevated MFN and down-regulated each of the 35 DUBs. To assess the effect of DUB silencing on steady-state MFN levels, we performed Western blotting analysis on cell lysates and quantified the levels of unmodified MFN normalized for the loading control and expressed it as fold change (Fig 1A). Flag-tagged MFN exhibited mitochondrial subcellular localization, and its expression in S2R+ cells resulted in an elongated mitochondrial network (Fig S1A). We identified two DUBs whose down-regulation resulted in decreased MFN levels (CG5798/USP8 and CG5384/USP14) and two DUBs, whose down-regulation resulted in increased MFN levels (CG5505/USP36, CG2904/Echinus) (Fig 1B). Down-regulation of Parkin or PINK1 increased MFN levels, as previously described (Tanaka et al, 2010; Ziviani et al, 2010). Of the two DUBs causing decreased MFN levels, USP8 was the highest scoring hit that decreased MFN levels (Fig 1B). USP8 interacts with many substrates such as the epidermal growth factor receptor, an essential regulator of proliferation and differentiation, and regulates endosomal trafficking by ubiquitin-mediated sorting of the endocytosed cargoes (Mizuno et al, 2005; Row et al, 2006; Williams & Urbe, 2007). Moreover, USP8 knockdown protects from α-synuclein–induced locomotor deficits and cell loss in an α-synuclein fly model of PD (Alexopoulou et al, 2016). It was also shown that USP8 regulates induced mitophagy by

controlling Parkin recruitment to depolarized mitochondria after CCCP treatment (Durcan et al, 2014). More recently, it has been found that it can regulate basal autophagy in the absence of CCCP, although its role has not been thoroughly characterized in this process and it is controversial (Jacomin et al, 2015). USP8 is also highly expressed in the brain and up-regulated in neurodegenerative conditions (Paiardi et al, 2014), which makes it of neurological interest.

### USP8 down-regulation correlates with decreased MFN protein levels

We next validated if USP8 down-regulation correlated with changes in MFN protein levels. Upon efficient USP8 down-regulation in fly cells, as assessed by qPCR (Fig S1B), steady-state levels of endogenous (Figs 1C and S1C) or exogenously expressed tagged MFN were decreased (Fig 1D) and mitochondria appeared accordingly fragmented (Fig S1D). The effect was specific for USP8 because re-expression of USP8 in USP8 RNAi cells restored MFN levels (Fig 1D). In contrast, in cells overexpressing USP8, levels of exogenously expressed (Fig 1D) and endogenous MFN were increased (Fig 1E) and mitochondria were elongated and clumped, accumulating in the perinuclear area (Fig S1E).

We next assessed the impact of USP8 down-regulation on MFN levels in vivo. To this aim, we drove efficient whole body USP8 knockdown (KD) by using the Actin5C driver (*Act*-GAL4>USP8-RNAi), achieving significant USP8 down-regulation at 29°C (Fig S1F). Attempts to increase USP8 down-regulation efficiency by using the stronger GAL4 driver *daughterless (da)* caused larvae lethality, suggesting that USP8 expression levels in vivo are tightly regulated. *Act*-GAL4>USP8-RNAi on the other hand was viable with no apparent locomotor defects. As previously observed in vitro, levels of MFN were reduced in vivo in USP8 down-regulating flies (Fig 1F). We also found decreased MFN levels in protein extracts coming from flies carrying heterozygous USP8 gene deletion (USP8$^{-/+}$) (Mukai et al, 2010), further supporting that the effect is specific for USP8 (Fig 1G).

### USP8 down-regulation ameliorates the phenotype of PINK1 KO flies

We addressed whether USP8 knockdown in PINK1 KO flies prevented the multiple phenotypes recapitulating key features of locomotor and cellular defects manifested in the flies as degeneration of dopaminergic (DA) neurons and reduced climbing ability. We also assessed the flight muscle, mitochondria ultrastructure, male fertility, and life span, all degenerated or affected in PINK1 KO flies (Clark et al, 2006; Park et al, 2006; Yang et al, 2006). Immunostaining for the specific DA neuronal marker tyrosine hydroxylase (TH) allowed the inspection of the DA neuronal network composed of well-characterized DA neuron clusters (PPM1, PPM2, PPM3, PPL1, PPL2, and VUM) in brains (Fig 2A). PINK1 KO showed the expected reduction in TH staining and exhibited a small but significant decrease in the number of DA neurons in the PPL1 DA neuronal cluster (Fig 2B and C) (Park et al, 2006; Wang et al, 2006; Yang et al, 2006). Accordingly, dopamine levels measured from PINK1 KO heads were significantly lower compared with control flies (Fig 2D). USP8 down-regulation completely prevented the loss of

**Table 1.** Complete list of the 75 human known or predicted DUBs and their fly homologue, when known or predicted, based on sequence similarity. Where available, Entrez/PubMed gene ID and fly gene name is provided.

| Gene name | Gene ID | Fly homologue | Fly gene name |
| --- | --- | --- | --- |
| UCHL1 | 7345 | CG4265 | |
| UCHL3 | 7347 | CG4265 | |
| BAP1 | 8314 | CG8445 | CALYPSO |
| UCHL5/UCH37 | 51377 | CG3431 | |
| DUB3 | 377630 | CG5505 | USP36/SCRAWNY |
| USP1 | 7398 | CG15817 | USP1 |
| USP2 | 9099 | CG14619 | |
| USP3 | 9960 | CG5798 | UBPY/USP8 |
| USP4 | 7375 | CG8334 | |
| USP5 | 8078 | CG12082 | |
| USP6 | 9098 | CG8334 | |
| USP7/HAUSP | 7874 | CG1490 | USP7 |
| USP8/USPY | 9101 | CG5798 | UBPY/USP8 |
| USP9X/FAM | 8239 | CG1945 | FAT FACETS |
| USP10 | 9100 | CG32479 | |
| USP11 | 8237 | CG8334 | |
| USP12 | 219333 | CG7023 | USP12-46 |
| USP13 | 8975 | CG12082 | USP5 |
| USP14 | 9097 | CG5384 | |
| USP15 | 9958 | CG12082 | |
| USP16 | 10600 | CG4165 | USP16-45 |
| USP18 | 11274 | CG5486 | USP64E/USP47 |
| USP19 | 10869 | CG8334 | |
| USP20 | 10868 | CG8494 | |
| USP21 | 27005 | CG14619 | |
| USP22 | 23326 | N/A | |
| USP24 | 23358 | CG1945 | FAT FACETS |
| USP25 | 29761 | CG5794 | PUF/USP34 |
| USP26 | 83844 | CG5798 | USP8/USPY |
| USP27X | 389856 | CG4166 | NOT |
| USP28 | 57646 | CG5794 | PUF/USP34 |
| USP29 | 57663 | CG5798 | USP8/USPY |
| USP30 | 84749 | CG3016 | |
| USP31 | 57478 | CG30421 | USP15-31 |
| USP32 | 84669 | CG8334 | |
| USP33 | 23032 | CG8494 | USP20-33 |
| USP34 | 9736 | CG5794 | PUF/USP34 |
| USP35 | 57558 | CG8830 | DUBAI |
| USP36 | 57602 | CG5505 | |
| USP37 | 57695 | CG5798 | USP8/USPY |
| USP38 | 84640 | CG8830 | DUBAI |

**Table 1.** Continued

| Gene name | Gene ID | Fly homologue | Fly gene name |
| --- | --- | --- | --- |
| USP39 | 10713 | CG7288 | |
| USP40 | 55230 | CG5486 | USP64E/USP47 |
| USP41 | 373856 | CG5486 | USP64E/USP47 |
| USP42 | 84132 | CG5505 | USP36/SCRAWNY |
| USP43 | 124739 | CG30421 | USP15-31 |
| USP44 | 84101 | CG5798 | USP8/USPY |
| USP45 | 85015 | CG4165 | USP16-45 |
| USP46 | 64854 | CG7023 | USP12-46 |
| USP47 | 55031 | CG5486 | USP64E/USP47 |
| USP48 | 84196 | CG1490 | USP7 |
| USP49 | 25862 | CG5798 | USP8/USPY |
| USP50 | 373509 | CG5798 | USP8/USPY |
| USP51 | 158880 | CG4166 | NOT |
| USP52 | 9924 | CG8232 | PAN2 |
| USP53 | 54532 | CG2904 | ECHINUS |
| USP54 | 159195 | CG2904 | ECHINUS |
| OTUB1 | 55611 | CG4968 | |
| CYLD | 1540 | CG5603 | |
| TNFAIP3/A20 | 7128 | CG9448 | TRABID |
| OTUD1 | 220213 | CG6091 | |
| YOD1 | 55432 | CG4603 | |
| OTUD3 | 23252 | CG6091 | |
| OTUD4 | 54726 | CG12743 | OTU |
| OTUD6A | 139562 | CG7857 | |
| OTUD6B | 51633 | CG7857 | |
| OTUD7A | 161725 | CG9448 | TRABID |
| OTUD7B | 56957 | CG9448 | TRABID |
| TRABID | 54764 | CG9448 | TRABID |
| ATXN3 | 4287 | CG13379 | |
| ATX3L | N.A. | CG13379 | |
| JOSD1 | 9929 | CG3781 | |
| JOSD2 | 126119 | CG3781 | |
| AMSH/STAMBP | 10617 | CG2224 | |
| AMSH-LP | 57559 | CG2224 | |

PINK1 KO DA neurons (Fig 2B and C), restoring dopamine to wild-type levels (Fig 2D). Moreover, USP8 down-regulation ameliorated the shorter longevity (Fig 2E), corrected thoracic muscle fiber disorganization with enlarged electron transparent mitochondria and irregular myofibril arrays (Park et al, 2006) (Fig 2F) typical of the PINK1 KO flies (Park et al, 2006). More importantly, ultrastructural transmission electron microscopy (TEM) analysis showed that the mitochondrial cristae, fragmented and sparely packed in PINK1 mutants, were recovered with highly increased electron-dense staining intensity (Fig 2G). USP8 knockdown also ameliorated the PINK1 climbing defect (Fig 2H).

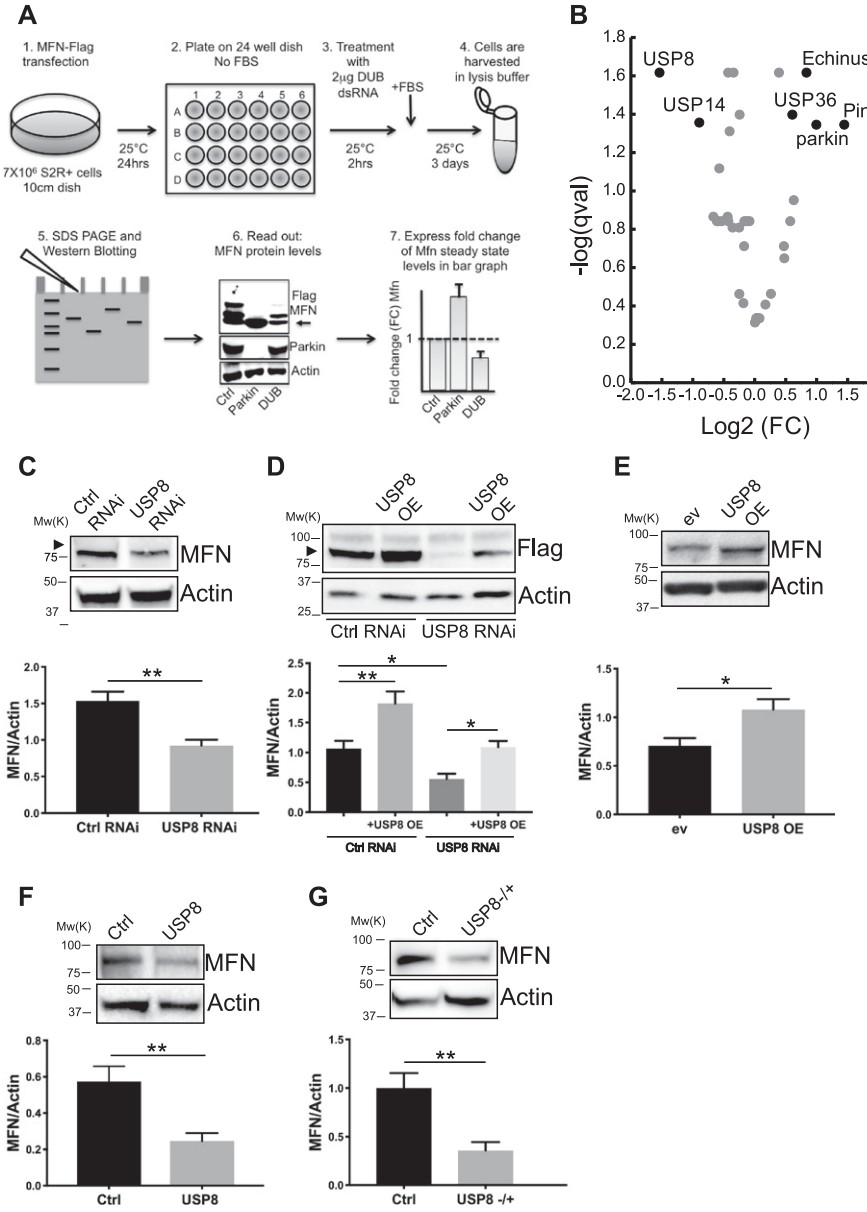

**Figure 1. A targeted siRNA screening identified DUB USP8 whose down-regulation correlates with decreased MFN levels.**

**(A)** siRNA screen to identify DUBs affecting pathologically elevated MFN protein levels. Protein extracts from *Drosophila* S2R+ cells expressing equal amounts of Flag-MFN and treated with 2 µg dsRNA probe were separated by SDS–PAGE and immunoblotted using an anti-Flag antibody. Densitometric analysis of MFN signal normalized to loading control and expressed as fold change (FC) versus control dsRNA was used as read out to identify DUBs whose down-regulation affects MFN protein levels. **(B)** Volcano plot constructed by plotting the negative log of the FDR corrected *P* value (qval) on the y-axis against the log of the FC calculated in (A). Those points that are found toward the top of the plot far to either the left- or right-hand side represent values with large FC and high statistical significance. A threshold of *P* < 0.05 and 0.75 < FC > 1.3 led to the identification of four DUBs whose down-regulation resulted in either decreased MFN levels (USP8 FC = 0.345 ± 0.04, qval = 0.024; USP14 FC = 0.537 ± 0.06, qval = 0.044) or increased MFN levels (Echinus FC = 1.784 ± 0.13, qval = 0.024; USP36 FC = 1.524 ± 0.12, qval = 0.040). Down-regulation of PINK1 or Parkin led to increased MFN levels (FC = 2.724 ± 0.44, qval = 0.045; and FC = 1.994 ± 0.28, qval = 0.045, respectively). **(C)** S2R+ cells were transfected with the indicated siRNA (Ctrl and USP8) and lysed after 3 d. Equal amounts of protein (30 µg) were separated by SDS–PAGE and immunoblotted using the indicated antibodies. Representative of n = 6. Graph bar shows mean ± SEM of ratio between densitometric levels of MFN and those of Actin from at least eight independent experiments. Means are significantly different according to *t* test; *P* = 0.0025 (**), n = 6. **(D)** S2R+ cells were transfected with the indicated plasmid (MFN-Flag, USP8) and siRNA (Ctrl and USP8) and lysed after 3 d. Equal amounts of protein (30 µg) were separated by SDS–PAGE and immunoblotted using the indicated antibodies. Representative of n = 5. Graph bar shows mean ± SEM of ratio between densitometric levels of Flag (MFN) and those of Actin relatively to control from at least four independent experiments. One-way ANOVA; *P* < 0.0001 (****), followed by Tukey's multiple comparison test. n = 5. **(E)** S2R+ cells were transfected with the indicated plasmids (empty vector, ev or USP8) and lysed after 3 d. Equal amounts of protein (30 µg) were separated by SDS–PAGE and immunoblotted using the indicated antibodies. Representative of n = 4. Graph bar shows mean ± SEM of ratio between densitometric levels of MFN and those of Actin relatively to control from at least four independent experiments. Means are significantly

different according to the *t* test; *P* = 0.0313 (*), n = 4. **(F)** Equal amounts of protein (70 µg), isolated from wild-type (Ctrl) flies or those down-regulating USP8 (USP8) separated by SDS–PAGE and immunoblotted using the indicated antibodies. Representative of n = 8. Graph bar shows mean ± SEM of ratio between densitometric levels of MFN and those of Actin relatively to control from at least three independent experiments. Means are significantly different according to the *t* test; *P* = 0.0044 (**), n = 8. The flies were raised at 29°C to allow efficient down-regulation of USP8. **(G)** Equal amounts of protein (70 µg), isolated from wild-type (Ctrl) flies and those carrying heterozygous deletion of USP8 (USP8-/+) separated by SDS–PAGE and immunoblotted using the indicated antibodies. Representative of n = 5. Graph bar shows mean ± SEM of ratio between densitometric levels of MFN and those of Actin relatively to control from at least four independent experiments. Means are significantly different according to the *t* test; *P* = 0.0069 (**), n = 5.

Source data are available for this figure.

To independently validate the previous results, we analyzed a bona fide genetic mutant for USP8. Heterozygous USP8 gene deletion (USP8$^{-/+}$) in PINK1 KO background also completely prevented the loss of DA neurons (Fig 3A and B), restored dopamine levels to wild-type (Fig 3C), corrected thoracic muscle fiber disorganization (Fig 3D) and mitochondrial structure (Fig 3E), ameliorated the shorter longevity (Fig 3F), and completely corrected the locomotor defects (Fig 3G). Thus, these observations support the specificity of

the previous results and confirm that loss of USP8 ameliorates PINK1 KO phenotypes.

## USP8 down-regulation rescues mitochondria defects of PINK1 KO flies

To verify if USP8 down-regulation also correlates to the amelioration of mitochondrial function, impaired in PINK1 KO/KD models

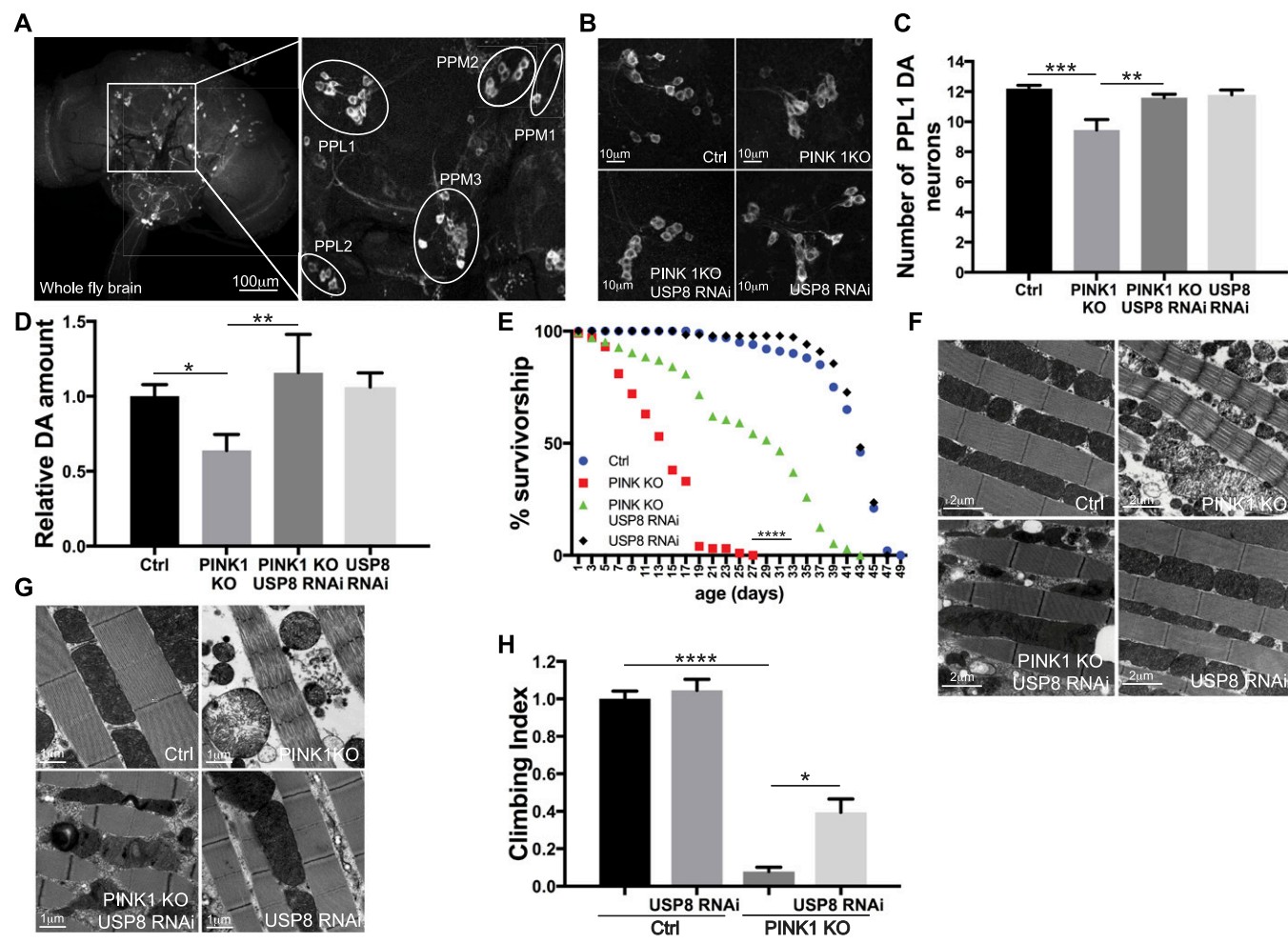

**Figure 2. USP8 down-regulation corrects DA neuron loss, life span, muscle degeneration, and locomotor impairment of PINK1-deficient flies.**
**(A)** Confocal images (projection, Z stack) of whole-mount adult brain (left panel) showing DA neuron clusters marked with an anti-TH antibody. Immunostaining for the specific DA neuronal marker TH allows the inspection of the DA neuronal network composed by well-characterized DA neuron clusters (PPM1, PPM2, PPM3, PPL1, PPL2, and VUM) in brains (right panel). **(B)** Whole brains of 15-d-old male flies of the indicated genotypes were immunostained with anti-TH antibody. Panel shows confocal images of PPL1 cluster DA neurons of the indicated genotypes. Representative of n = 15. **(C)** Bar graph shows the number of DA neurons in the PPL1 cluster of the brains of the indicated genotypes. One-way ANOVA, $P < 0.0001$ (****) followed by Tukey's multiple comparison test; n = 15. **(D)** Relative dopamine amount from 15-d-old adult heads of the indicated genotype normalized to control flies. One-way ANOVA, $P = 0.0073$ (**) followed by Tukey's multiple comparison test. n = 4. **(E)** Life span analysis of adult males of the indicated genotypes. Male flies of the indicated genotypes were collected during 12 h after hatching and transferred to fresh food every 2 d and dead flies were counted in the same interval. At least 100 flies per genotype were used for the analysis. Log-rank, Mantel–Cox test (Ctrl versus PINK1 KO $P < 0.0001$; Ctrl versus PINK1 KO USP8 RNAi $P < 0.0001$; Ctrl versus USP8 RNAi $P > 0.05$; PINK1 KO versus PINK1 KO USP8 RNAi $P < 0.0001$; PINK1 KO versus USP8 RNAi $P < 0.0001$; and PINK1 KO USP8 RNAi versus USP8 RNAi $P < 0.0001$ $P < 0.0001$). **(F)** Ultrastructural analysis of the indirect flight muscles from fly thoraces of the indicated genotypes. Images show TEM images of thorax muscles from flies of the indicated genotypes. Representative of n = 3. **(G)** Enlarged TEM images of flight muscle mitochondria of the indicated genotypes. Representative of n = 3. **(H)** Graph bar shows mean ± SEM of the climbing performance of flies of the indicated genotype from at least three independent experiments. One-way ANOVA, $P < 0.0001$ (****); Tukey's multiple comparison test; n = 3.

(Clark et al, 2006; Gandhi et al, 2009; Morais et al, 2014; Park et al, 2006), we measured mitochondrial respiration in digitonin-permeabilized cells, where mitochondria are directly accessible to substrates. In line with what has been previously reported (Gandhi et al, 2009; Morais et al, 2009), we found that ADP-stimulated glutamate-supported respiration (state 3) was significantly reduced in cells lacking PINK1 (Fig S2A). State 3/basal (state 4) respiration ratio, also known as respiratory control ratio (RCR), was reduced (Fig S2B). USP8 down-regulation did not perturb mitochondrial respiration per se; however, it corrected the respiration defects of the PINK1-deficient cells (Fig S2A and B). In cells

lacking PINK1, mitochondrial dysfunction is mirrored also by changes in mitochondrial membrane potential ($\Delta\psi_m$) (Gandhi et al, 2009; Morais et al, 2009; Mortiboys et al, 2008). When we measured latent mitochondrial dysfunction using a well-established assay based on the response of $\Delta\psi_m$ to the ATPase inhibitor oligomycin, as expected (Gandhi et al, 2009; Morais et al, 2009), we noticed that PINK1-deficient mitochondria sustain their $\Delta\psi_m$ by hydrolyzing cytosolic ATP and therefore depolarize after oligomycin treatment (Fig S2C–E). Although down-regulation of USP8 had no effect on $\Delta\psi_m$ in PINK1-deficient cells, it fully prevented the oligomycin-induced depolarization, further confirming its beneficial effects on

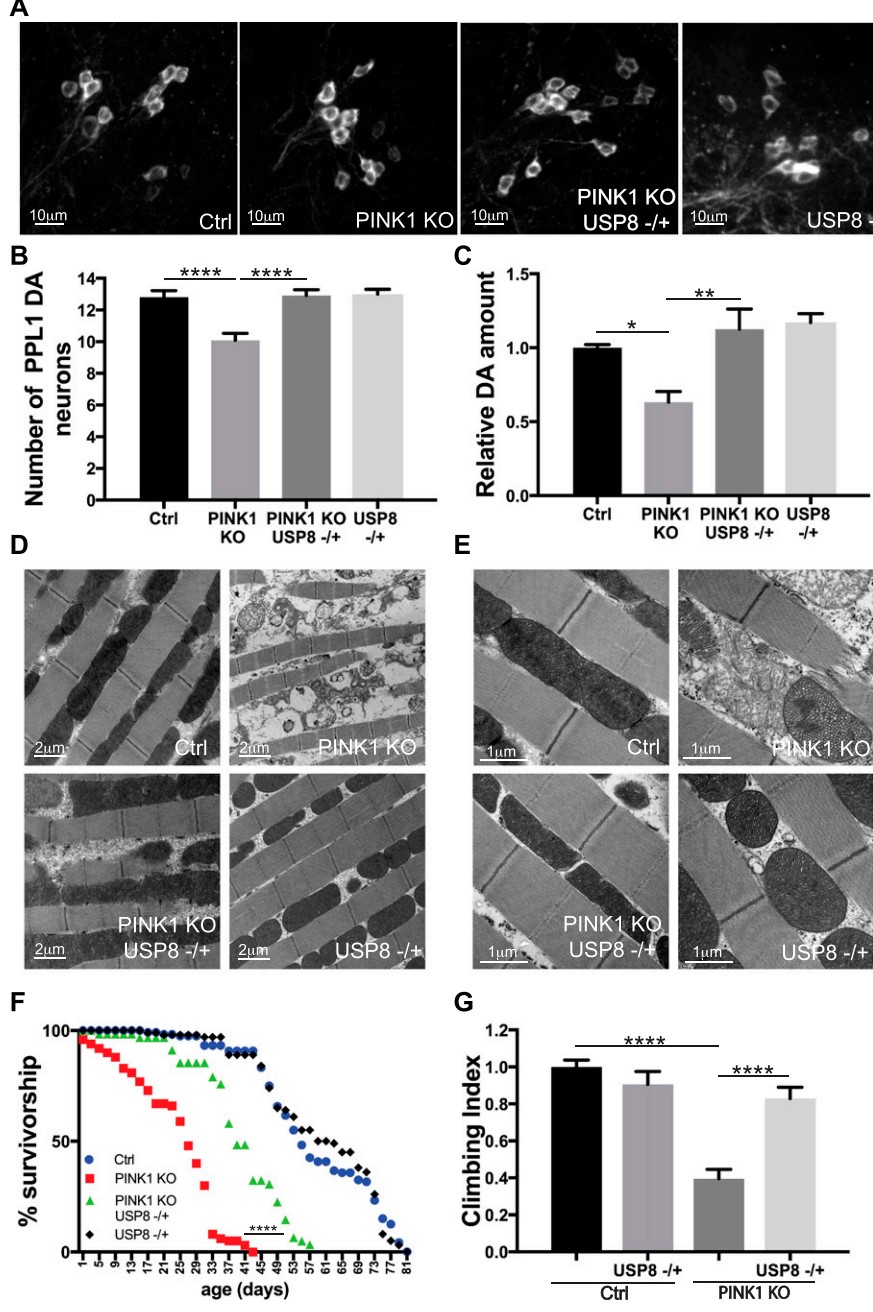

**Figure 3. Heterozygous USP8 gene deletion corrects DA neuron loss, life span, muscle degeneration, and locomotor impairment of PINK1-deficient flies.**
**(A)** Whole brains of 15-d-old male flies of the indicated genotypes were immunostained with anti-TH antibody. Panel shows confocal images of DA neuron of the PPL1 cluster of the indicated genotypes. Representative of n = 11. **(B)** Bar graph shows the number of DA neurons in the PPL1 cluster of the brains of the indicated genotypes. One-way ANOVA, $P < 0.0001$ (****); Tukey's multiple comparison test; n = 11. **(C)** Relative dopamine amount from 15-d-old adult heads of the indicated genotype normalized to control flies. One-way ANOVA, $P = 0.0002$ (***); Tukey's multiple comparison test; n = 5. **(D)** TEM images of thorax muscles from flies of the indicated genotypes. Representative of n = 3. **(E)** Enlarged TEM image of flight muscle mitochondria of the indicated genotypes. Representative of n = 3. **(F)** Life span analysis of adult males of the indicated genotypes. Male flies of the indicated genotypes were collected during 12 h after hatching and transferred to fresh food every 2 d and dead flies were counted in the same interval. At least 100 flies per genotype were used for the analysis. Log-rank, Mantel–Cox test (Ctrl versus PINK1 KO $P < 0.0001$; Ctrl versus PINK1 KO USP8–/+ $P < 0.0001$; Ctrl versus USP8–/+ $P > 0.05$; PINK1 KO versus PINK1 KO USP8–/+ $P < 0.0001$; PINK1 KO versus USP8–/+ $P < 0.0001$; and PINK1 KO USP8–/+ versus USP8–/+ $P < 0.0001$ $P < 0.0001$). **(G)** Graph bar shows mean ± SEM of the climbing performance of flies of the indicated genotype from at least three independent experiments. One-way ANOVA, $P < 0.0001$ (****); Tukey's multiple comparison test; n = 3.

mitochondrial function (Fig S2C–E). Because USP8 participates in a multiplicity of pathways (Alexopoulou et al, 2016; Durcan & Fon, 2015; Mizuno et al, 2005; Row et al, 2006), the beneficial effects on mitochondrial function measured in situ might be indirect. We, therefore, compared the function of mitochondria purified from PINK1-mutant (KO) flies with that recorded in mitochondria isolated from PINK1 KO flies where we down-regulated USP8 (Fig 4A and B) or from double heterozygous USP8-deficient (USP8$^{-/+}$), PINK1 KO flies (Fig 4C and D). As expected, glutamate-supported ADP-stimulated respiration was reduced, resulting in lower RCR in isolated PINK1 KO mitochondria (Gandhi et al, 2009; Morais et al, 2009) (Fig 4A–D). On the other hand, USP8 RNAi (Fig 4A and B) or heterozygous USP8 gene deletion (Fig 4C and D) in PINK1 KO flies normalized ADP-stimulated respiration and RCR.

Blue Native PAGE (BN-PAGE) of mitochondrial extracts lent further biochemical support to the measured functional amelioration. Extracts from PINK1-deficient flies displayed reduced levels of respiratory complex I, which was corrected by heterozygous deletion of USP8 (Fig 4E and F). PINK1 mutants show reduced enzymatic activity of complex I (Morais et al, 2014; Pogson et al, 2014). Both USP8 fly lines (USP8$^{+/-}$ and USP8 RNAi) restored complex I activity of PINK1 mutants (Fig 4G and H).

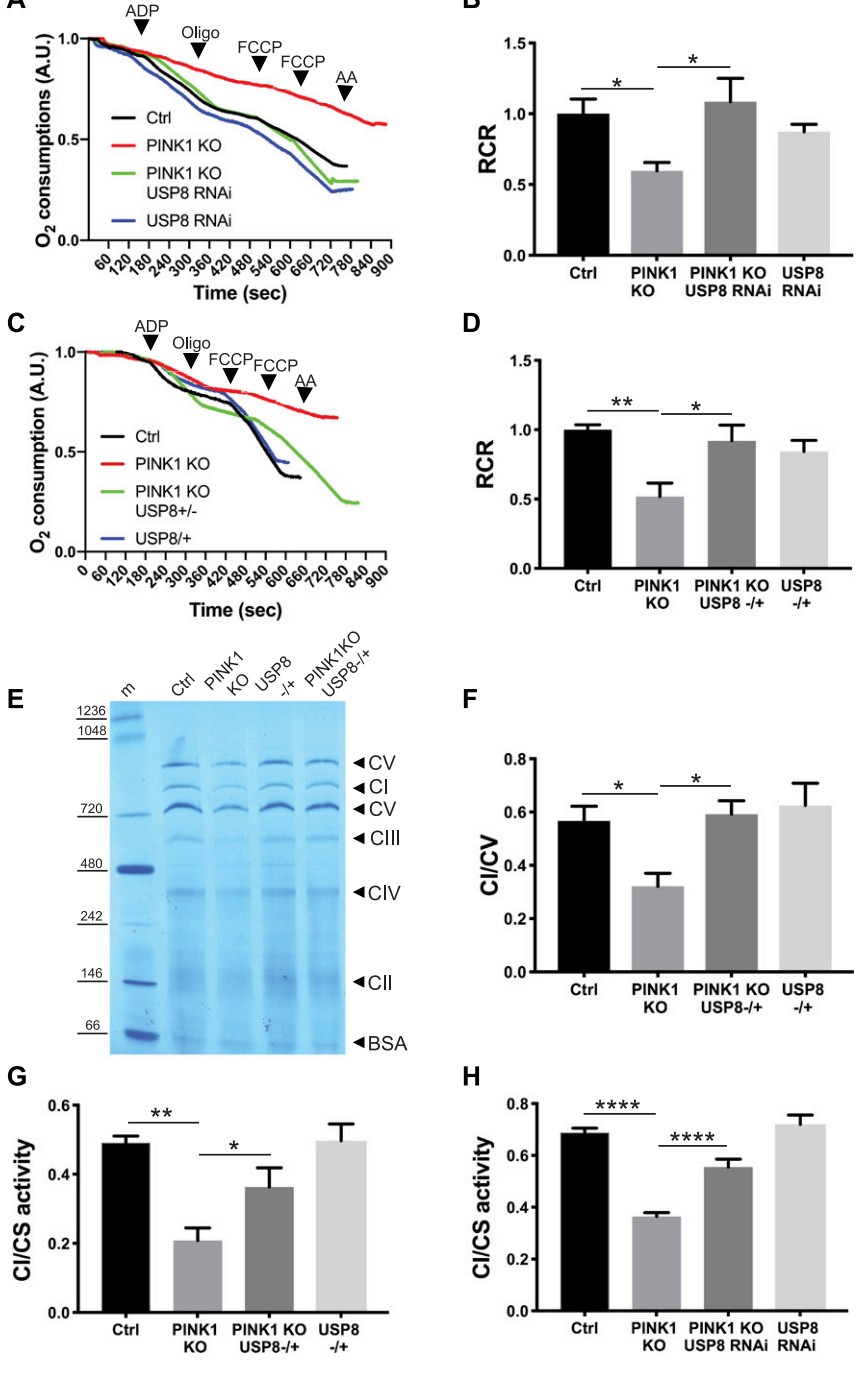

**Figure 4. USP8 down-regulation rescues PINK1-deficient mitochondria respiratory defects ex vivo.**
**(A)** Representative traces of oxygen consumption of intact isolated mitochondria extracted from flies of the indicated genotype and subjected to 10 mM/5 mM pyruvate/malate 200 μM ADP, 2 μg/ml oligomycin, and 200 nM FCCP, 2 μM antimycinA, respectively. Representative of n = 5. **(B)** Quantitative analysis of respiratory fitness of isolated mitochondria extracted from flies of the indicated genotype treated as in (A). Graph shows mean ± SEM (n = 5 independent experiments) of RCR relative to ctrl. One-way ANOVA, P = 0.0074 (**); Tukey's multiple comparison test; n = 5. **(C)** Representative traces of oxygen consumption of intact isolated mitochondria extracted from flies of the indicated genotype and subjected to 10 mM/5 mM pyruvate/malate 200 μM ADP, 2 μg/ml oligomycin, and 200 nM FCCP, 2 μM antimycinA, respectively. Representative of n = 5. **(D)** Quantitative analysis of respiratory fitness of isolated mitochondria extracted from flies of the indicated genotype treated as in (G). Graph shows mean ± SEM (n = 5 independent experiments) of RCR relative to ctrl. One-way ANOVA, P = 0.0064 (**); Tukey's multiple comparison test; n = 5. **(E)** Blue Native PAGE of mitochondrial extracts from flies of the indicated genotypes. Respiratory complexes were separated in a non-denaturing polyacrylamide gel. Representative of n = 3. **(F)** Densitometric analysis of (E). Graph bar shows mean ± SEM of ratio between densitometric levels of complex I (CI) and those of complex V (CV). One-way ANOVA, P = 0.0282 (**); Tukey's multiple comparison test; n = 3. **(G)** Graph shows mean ± SEM (n = 4 independent experiments) of complex I activity relatively to citrate synthase (CS) activity in isolated 2.5 μM alamethicin-treated mitochondria extracted from flies of the indicated genotype. One-way ANOVA, P = 0.0012 (**); Tukey's multiple comparison test; n = 4. **(H)** Graph shows mean ± SEM (n = 7 independent experiments) of complex I activity relatively to CS activity in isolated 2.5 μM alamethicin-treated mitochondria extracted from flies of the indicated genotype. One-way ANOVA, P < 0.0001 (****); Tukey's multiple comparison test; n = 7.

PINK1-null mutant males are sterile, as a consequence of spermatogenesis defects deriving from mitochondrial dysfunction (Clark et al, 2006; Deng et al, 2008; Greene et al, 2003; Park et al, 2006). Of note, heterozygous USP8 gene deletion favours the restoring of sperm production of the PINK1 KO, rescuing male sterility (Fig S3). The seminal vesicles of Ctrl and USP8−/+ males were well developed, swollen, and brownish in color (Fig S3A and B), whereas those of PINK1 KO were reduced in volume and more transparent (Fig S3C). Puncturing the vesicles of Ctrl and USP8−/+ males released

a large amount of sperm (Fig S3E and F), whereas sperm was almost absent in PINK1 KO vesicles (Fig S3G). Rescued males (PINK1 KO, USP8−/+) showed an intermediate pattern, with swollen, opaque vesicles (Fig S3D) releasing some sperm groups (Fig S3H). The fluorescence staining revealed a difference among the four male groups also in the accessory glands' wall, whose cells appeared alive (green) in ctrl and USP8−/+ males (Fig S3I and J) and dead (red) in PINK1 KO (Fig S3K). In rescued males (PINK1 KO, USP8−/+), part of the accessory glands' cells was alive (Fig S3L). The result of

fluorescence staining proves that the effect is not limited to sperm production, but it is also extended to the functionality of the accessory glands, that play a crucial role on both male fertilization success and female fertility (Simmons & Fitzpatrick, 2012).

Taken together, these analyses show that the mitochondrial-defective phenotype of PINK1 KO flies can be recovered by decreasing USP8 expression, including complex I levels and activity.

### Pharmacological inhibition of USP8 corrects PINK1-deficient flies

The genetic experiments showed that USP8 inhibition ameliorates all the phenotypes that we tested that are associated to *Drosophila* PINK1 KO. We, therefore, decided to test in vivo the effect of DUBs-IN-2 (ChemScene LLC), a potent and membrane-permeant USP8 drug inhibitor. DUBs-IN-2 is highly selective for USP8 with a half maximal inhibitory concentration ($IC_{50}$) of 0.28 $\mu$M (Colombo et al, 2010) and small or no effect on USP7 ($IC_{50}$ > 100 $\mu$M for USP7). The compound has been described as an inhibitor of human USP8, which shares about ~45% sequence homology to the fly ortholog. DUBs-IN-2 was mixed in the fly food

with the food-coloring patent blue V (E131) to monitor drug ingestion (Fig S4A). Increasing inhibitor concentrations did not affect the food uptake of flies as measured by E131 absorbance in fly lysates (Fig S4B) and did not affect locomotor behavior in a control background (Fig S4C). Remarkably, DUBs-IN-2 administered to adult PINK1-deficient flies significantly suppressed the locomotor deficits (Fig 5A). Dose–response curve indicated the best rescue of PINK1 KO climbing performance upon 10 $\mu$M DUBs-IN-2 administration (Fig S4C). DUBs-IN-2 administration to PINK1 KO flies also prevented loss of DA neurons (Fig 5B and C), restored dopamine levels (Fig 5D), and it modestly ameliorated longevity (Fig 5E).

### USP8 down-regulation corrects pathologically elevated MFN levels of PINK1 and Parkin KO flies

PINK1 loss-of-function results in increased MFN protein levels (Tanaka et al, 2010; Ziviani et al, 2010), altered mitochondrial morphology (Mortiboys et al, 2008; Narendra et al, 2008; Tanaka et al, 2010; Ziviani et al, 2010), impaired mitophagy (Gegg et al, 2010;

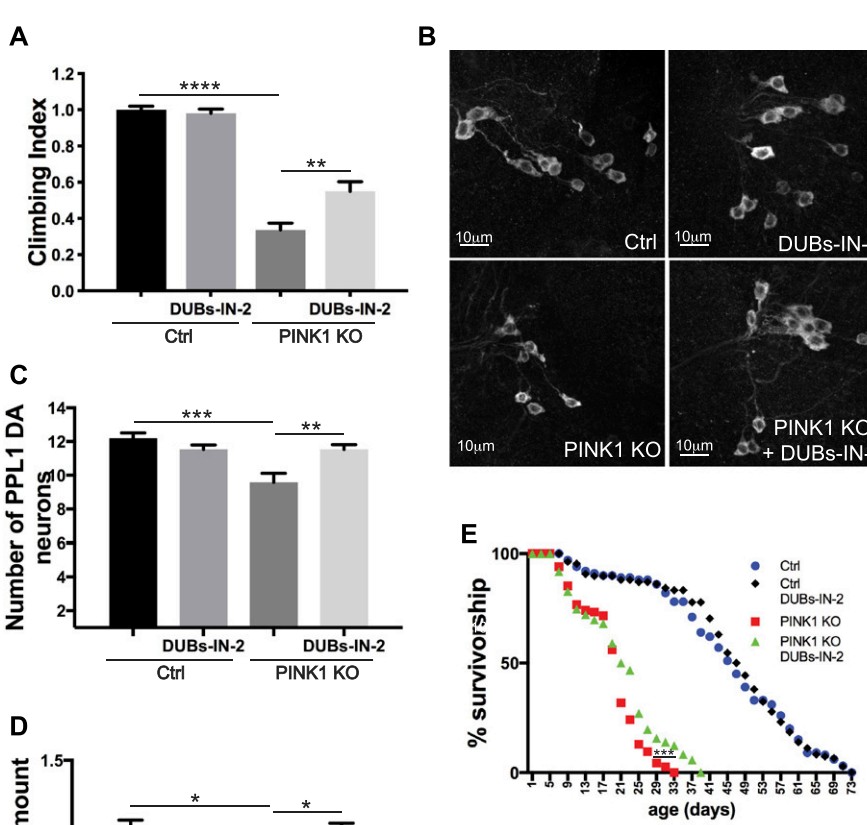

**Figure 5. Pharmacological USP8 inhibition corrects DA neuron loss, life span, muscle degeneration, and locomotor impairment of PINK1-deficient flies.**
**(A)** Graph bar shows mean ± SEM of the climbing performance of 3-d-old flies of the indicated genotype or treated with DUBs-IN-2 or DMSO for 48 h from at least four independent experiments. Two-way ANOVA *P* < 0.0001 (****); Tukey's multiple comparison test; n = 8. **(B)** Whole brains of 15-d-old male flies of the indicated genotypes or treated with DUBs-IN-2 for 15 d were immunostained with anti-TH antibody. Panel shows (projection, Z stack) confocal images of PPL1 cluster DA neurons of the indicated genotypes. Representative of n = 9. **(C)** Bar graph shows the number of PPL1 cluster DA neurons in brains of the indicated genotypes treated with DUBs-IN-2 or DMSO for 15 d. Two-way ANOVA, *P* = 0.0004 (***); Tukey's multiple comparison test; n = 15. **(D)** Relative dopamine amount from 15 d old adult heads of the indicated genotype treated with DUBs-IN-2 or DMSO for 15 d normalized to control flies. Two-way ANOVA *P* = 0.0217 (*); Tukey's multiple comparison test; n = 3. **(E)** Life span analysis of male flies of the indicated genotypes treated with DUBs-IN-2 or DMSO. At least 100 flies were used for the analysis. Log-rank, Mantel–Cox test (Ctrl versus PINK1 KO *P* < 0.0001; Ctrl versus PINK1 KO+DUBs-IN-2, *P* < 0.0001; Ctrl versus Ctrl+DUBs-IN-2 *P* > 0.05; PINK1 KO versus PINK1 KO+DUBs-IN-2 *P* < 0.001; PINK1 KO versus Ctrl+DUBs-IN-2 *P* < 0.0001; and PINK1 KO+DUBs-IN-2 versus Ctrl+DUBs-IN-2 *P* < 0.0001).

Narendra et al, 2008; Ziviani et al, 2010), and oxidative phosphor-ylation (Morais et al, 2009, 2014), with mitochondrial Ca$^{2+}$ overload and increased reactive oxygen species production (Gandhi et al, 2009). Similar phenotypes are caused by altered MFN, which prompted us to investigate whether USP8 down-regulation corrected pathologically elevated MFN levels of PINK1 KO flies. Indeed, USP8 down-regulation in vivo completely normalized increased MFN levels of PINK1 KO (Fig 6A). Pharmacological inhibition of USP8 also led to reduced PINK1 KO MFN protein levels in flies, indicating that the inhibitor phenocopied genetic inhibition of USP8 (Fig 6B). Like PINK1, Parkin KO/KD also results in increased MFN protein levels (Gegg et al, 2010; Tanaka et al, 2010; Ziviani et al, 2010). We, therefore, assessed the effect of USP8 KD in a Parkin loss-of-function model of pathologically elevated MFN levels. USP8 KD corrected elevated MFN levels of Parkin KO flies (Fig 6C). It also recovered the disorganized muscle fibers with irregular arrangement of myofibrils and the swollen mitochondria of Parkin flies

(Fig 6D), and normalized the number of DA neurons that are decreased in Parkin KO background (Fig 6E). Interestingly, USP8 KD or inhibition did not correct climbing defects in Parkin KO flies (Fig 6F), nor in PINK1:Parkin double KO (Fig 6G).

# Discussion

Interventions that decrease MFN levels in PINK1 or Parkin KO flies can ameliorate the multiple phenotypes that are associated with the KO backgrounds (Deng et al, 2008; Poole et al, 2008; Liu et al, 2011; Vilain et al, 2012; Celardo et al, 2016). We, therefore, conducted an RNAi-based screening to identify DUBs that regulate MFN protein levels. We found USP8, a DUB previously identified in the regulation of endosomal trafficking (Mizuno et al, 2005; Row et al, 2006), CCCP-induced mitophagy (Durcan et al, 2014) and basal autophagy (Jacomin et al, 2015), and which down-regulation is protective from

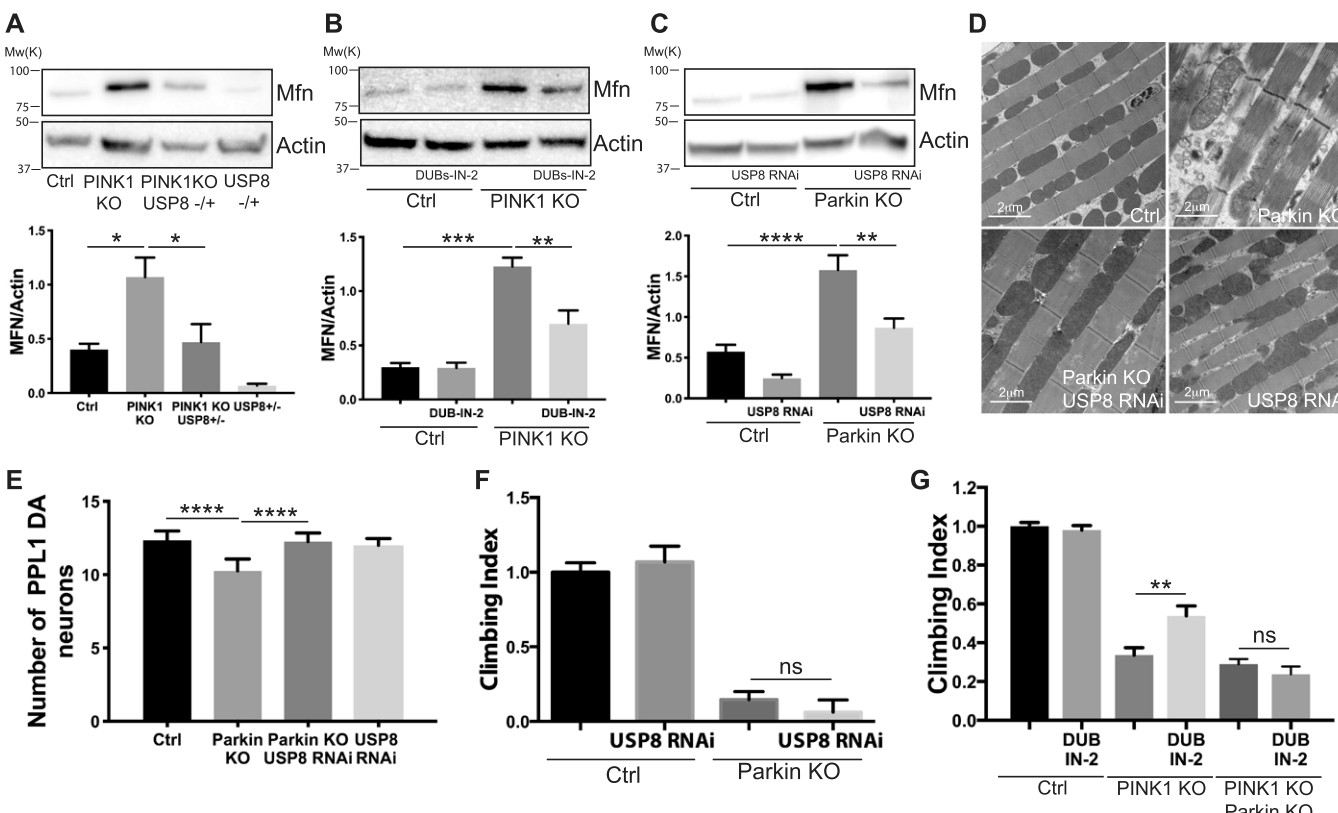

**Figure 6. USP8 down-regulation corrects pathologically elevated MFN levels of PINK1 and Parkin KO flies.**
**(A)** Equal amounts of protein (70 μg), isolated from flies of the indicated phenotype were separated by SDS–PAGE and immunoblotted using the indicated antibodies. Representative of n = 6. Graph bar shows mean ± SEM of ratio between densitometric levels of MFN and those of Actin from at least six independent experiments. One-way ANOVA, P = 0.0003 (***); Tukey's multiple comparison test; n = 6. **(B)** Equal amounts of protein (70 μg), isolated from flies treated with DUBs-IN-2 or DMSO for 48 h were separated by SDS–PAGE and immunoblotted using the indicated antibodies. Representative of n = 3. Graph bar shows mean ± SEM of ratio between densitometric levels of MFN and those of Actin from at least three independent experiments. Two-way ANOVA, P < 0.00001 (****); Tukey's multiple comparison test; n = 3. **(C)** Equal amounts of protein (70 μg), isolated from flies of the indicated phenotype were separated by SDS–PAGE and immunoblotted using the indicated antibodies. Representative of n = 9. Graph bar shows mean ± SEM of ratio between densitometric levels of MFN and those of Actin from at least nine independent experiments. One-way ANOVA, P < 0.0001 (****); Tukey's multiple comparison test; n = 9. **(D)** TEM images of thorax muscles from flies of the indicated genotypes. Thoraces were dissected from 3-d-old adult flies and fixed in 2% paraformaldehyde and 2.5% gluteraldehyde. The samples were rinsed, dehydrated, and embedded using Epon. Ultrathin sections were examined using TEM. Representative of n = 3. **(E)** Bar graph shows the number of DA neurons in the PPL1 cluster of the brains of the indicated genotypes. One-way ANOVA, P < 0.0001 (****); Tukey's multiple comparison test; n = 10. **(F)** Graph bar shows mean ± SEM of the climbing performance of flies of the indicated genotype from at least three independent experiments. One-way ANOVA, P < 0.0001 (****); n = 3. **(G)** Graph bar shows mean ± SEM of the climbing performance of flies of the indicated genotype from at least three independent experiments. Two-way ANOVA, P < 0.0001 (****); Tukey's multiple comparison test; n = 7.

α-synuclein–induced locomotor deficits in flies (Alexopoulou et al, 2016). Our data show that inhibition of USP8 in vitro and in vivo correlated with decreased mitochondrial fusion protein MFN, one of the bona fide Parkin targets (Gegg et al, 2010; Poole et al, 2010; Tanaka et al, 2010; Ziviani et al, 2010; Sarraf et al, 2013) (Fig 1), ameliorated PINK1 KO phenotypes in vivo (Figs 2, 3, and 5) and PINK1 KO mitochondrial dysfunction (Fig 4), and corrected MFN protein levels, increased in PINK1 KO models (Fig 6). Interestingly, USP8 KD also corrected MFN protein levels of Parkin KO flies, indicating that the effect on the levels of MFN is Parkin independent. USP8 KD also prevented Parkin KO DA neurons loss and normalized mitochondrial morphological defects, although it did not ameliorate Parkin climbing performance (Fig 6).

It has been shown that the knockdown of MFN is able to rescue the mitochondrial defects and the overall phenotypes of Drosophila PINK1 KO flies (Deng et al, 2008; Poole et al, 2008). More recently, it was shown that MFN knockdown can suppress loss of DA neurons of the PPL1 cluster and thorax deformation resulting from crushed thoracic muscle of the PINK1 KO flies, but not the mitochondrial defects (Celardo et al, 2016). We found that normalizing MFN levels of PINK1 KO flies by driving efficient whole body MFN KD (Debattisti et al, 2014) ameliorated the disorganized muscle fibers and mitochondria ultrastructure of PINK1 KO flies, but dopamine content and climbing performance were only modestly recovered, even if MFN levels of PINK1 KO flies were completely corrected (Fig S5). This result indicates that MFN normalization deriving from USP8 KD likely contributes to the amelioration of the PINK1 phenotype but does not explain the full recovery of the multiple phenotypes that are associated with PINK1 loss. Indeed, our in vivo analysis indicates that USP8 KD has a broader protective effect than MFN KD and unlike MFN KD (Celardo et al, 2016; Vilain et al, 2012), it correlates with full correction of mitochondrial respiratory defects, complex I content and activity, and mitochondrial membrane potential of PINK1 KO flies (Figs 4 and S2). Previous examination of the PINK1-mutant phenotype demonstrated that although decreasing mitochondrial fusion rescues morphological mitochondrial defects of PINK1 flies, manipulation of mitochondrial fusion (or fission) does not rescue other PINK1-related phenotypes such as the reduced activity of complex I, loss of mitochondrial membrane potential, ATP content, and defective neurotransmitter release (Vilain et al, 2012; Vos et al, 2012). In light of this, we hypothesize that the protective effect of USP8 inhibition comes from a combination of signaling pathways, which directly or indirectly impinges on MFN levels and mitochondrial function. In mammals, it is established that USP8 is involved in endosomal trafficking (Clague et al, 2013), although its activity can have opposing effects. For instance, deubiquitination by USP8 was reported to slow the degradation of substrates (Mizuno et al, 2005; Mukai et al, 2010), but also to facilitate endosomal trafficking and lysosomal degradation (Row et al, 2007; Ali et al, 2013). shRNA against USP8 in SH-SY5Y neuroblastoma cells promotes α-synuclein degradation by the lysosome, which exerts a protective effect in vivo in an α-synuclein fly model of PD (Alexopoulou et al, 2016). It was also reported that USP8 is required for lysosomal biogenesis and productive autophagy in Drosophila larval fat body but inhibits basal autophagy in vitro in HeLa cells (Jacomin et al, 2015). Finally, deubiquitination of Parkin by USP8 is required for Parkin recruitment to CCCP-intoxicated

mitochondria and to promote stress-induced mitophagy in vitro (Durcan et al, 2014). Thus, USP8 down-regulation in this context inhibits Parkin recruitment to mitochondria, causing a delay in mitochondria clearance by mitophagy. In light of these seemingly opposing phenotypic outcomes, it is clear that USP8 has pleiotropic effects that depends on the specific genetic repertoire of the cell/tissue, varies in response to physiological versus pathological conditions, or might simply operate differently in cell lines versus the whole organism. Our model is consistent with a role of USP8 in controlling mitochondrial function via Parkin-independent regulation of pathologically elevated MFN protein levels. Yet, it does not exclude MFN-unrelated pathways that nevertheless impinge on mitochondrial function via Parkin, like the mitochondrial-derived vesicle pathway regulating mitochondria quality control (McLelland et al, 2014), or the endosomal–lysosomal pathway that can also play a role in selective degradation of dysfunctional mitochondria (Hammerling et al, 2017a, 2017b). Interestingly, it was shown in the latter that the autophagic activity is increased when the endosomal activity is impaired, sustaining the hypothesis that there is crosstalk between the various degradation pathways to ensure effective clearance. It is tempting to hypothesis an enhancement of autophagy deriving from USP8 KD to complement for impaired endosomal-mediated quality control. For these reasons, future studies need to be conducted in vivo to validate this hypothesis and clearly dissect coordination and timing of activation of these pathways in different tissues under physiological and pathological conditions.

Because of their involvement in the regulation of important signaling pathways, DUBs are emerging as extremely attractive druggable candidates (Sugiura et al, 2013). In recent years, many DUBs emerged as therapeutic targets to compensate for impaired mitophagy in PD (Bingol et al, 2014; Cornelissen et al, 2014; Wang et al, 2015; Chakraborty et al, 2018). Mitophagy is triggered by ubiquitin modification of mitochondrial proteins, which is in principle subject to suppression by deubiquitination. It is, therefore, reasonable that inhibition of specific DUBs should induce mitophagy and that it does so by deubiquitination mitochondrial proteins. Clinical trials for specific inhibitors of the ubiquitin–proteosome system have already been approved in cancer therapy for the treatment of multiple myeloma (Colland, 2010). Moreover, high-throughput screening of small chemical libraries identified non-selective DUB inhibitors as potent inducers of apoptosis in various cancer cells (Liu et al, 2003; Brancolini, 2008; Engels et al, 2009; Hussain et al, 2009; Py et al, 2013). Similarly, specific DUB inhibitors (or activators) can affect cellular response to stimuli that induce cell death. In this respect, the identification of a specific DUB that normalizes mitochondrial function might be instrumental to develop specific isopeptidase inhibitors that can modulate the fundamental biological process of mitochondria physiology and fitness, supporting the potential of USP8 inhibitors as therapeutics.

## Online methods

### Cell culture and transfection

Drosophila S2R+ cells were cultured in Schneider's medium (Invitrogen) supplemented with 10% heat-inactivated fetal calf serum (Sigma-Aldrich). The cells were maintained at 25°C and

passaged routinely before they reached confluence, to maintain a logarithmic growth. The cells were transfected using TransFectin lipid reagent (Bio-Rad) or Effectene (QIAGEN) following the manufacturer's instructions. In brief, 0.6 million cells were plated in six-well plate and transfected with 2 µg DNA/5 µl TransFectin or 1 µg DNA/10 µl Effectene + 8 µl Enhancer, 1 d after plating. The cells were collected 24–48 h after transfection. 500 µM copper sulfate solution was added to the cells to induce plasmid expression when required.

### Plasmids

MitoDsRed was subcloned from pDsRed2-Mito vector (Clontech) into pAct-PPA expression plasmid. C-terminal Flag tag MFN was obtained by amplification from cDNA clone (RE04414) and subcloned into pAct-PPA expression plasmid. CG5798/USP8 was amplified from cDNA clone and subcloned into pMt copper-inducible vector (Invitrogen).

### Gene silencing

*Drosophila* dsRNA probes were prepared using MEGA script kit (Ambion) following the manufacturer's instructions. The following primers have been used to prepare the RNAi probes: PINK1 CAATGTGACTTCTCCAGCGA and TCGTAGCGTTTCATCAGCAG; Parkin CTGTTGCAATTTGGAGGGA and CTTTGGCACGGACTCTTTCT; and MFN GGAACCTCTTTATTCTCTAT and GGTTTGCTTTGCCCCAACAT. CG5798/USP8 dsRNA probe was acquired from the Sheffield RNAi Screening Facility. 1.2 millions cells were plated on a six-well plate and treated with 7 µg RNAi probe in serum-free medium. 2 h after the probe treatment, complete medium was added to the wells, and the cells were cultured for 2 d before being transfected with indicated fly expression plasmids as previously described.

### Immunoblotting

Western blotting was performed using standard techniques. In brief, the cells were collected in lysis buffer (50 mM Tris–HCl, pH 8, 150 mM HCl, 1 mM MgCl$_2$, 2 mM EGTA, 1% Triton X, 10% glycerol, 10 mM NEM, 10 µM MG132 and protease inhibitor cocktail by Roche) and incubated on ice for 30 min before being centrifuged at maximum speed at 4°C. Ten to twelve flies were homogenized using a mortar and pestle in protein extraction buffer (200–300 µl, 150 mM NaCl, 5 mM EDTA, pH 8.0, 50 mM Tris, pH 8.0, 1% NP-40, 0.1% SDS 0.1, supplemented with 10 µM MG132, 10 mM NEM, and protease inhibitor cocktail). The following commercial antibodies were used: anti-Flag (1:1,000; Cell Signaling Technology), anti-Actin (1:10,000; Chemicon) has been described before. Anti-Drosophila Mitofusin (1:2000) was raised in rabbit against an N-terminal peptide, DTVDKSGPGSPLSRF. For detection, secondary antibodies conjugated with HRP (Chemicon) were used (1:3,000), and immunoreactivity was visualized with ECL chemiluminescence (Amersham).

### Live imaging

Cells were grown on imaging dishes (Chamber Slide Lab-Tek II 8; Thermo Fisher Scientific) or coverslips. After appropriate treatment, when indicated, the cells were treated with the selective mitochondrial dye Mitotracker (50 nM; Molecular Probe) for 10 min, washed three times with PBS, and imaged live in growing medium under ambient conditions on an Andromeda iMIC spinning disk live cell microscope with confocal resolution (TILL Photonics, 60X objective). For confocal z-axis stacks, 40 images separated by 0.2 µm along the z-axis were acquired.

For measurements of mitochondrial membrane potential, the cells were loaded with 25 nM tetramethylrhodamine methyl ester (TMRM) for 30 min at room temperature, and the dye was present during the experiment together with the multidrug resistance inhibitor cyclosporine H (1 µM). The cells were then observed using an Olympus IX81 inverted microscope equipped with a cell imaging system. Sequential images of TMRM fluorescence were acquired every 60 s with a 40× objective (Olympus). Where indicated, oligomycin (2.5 µg/ml; Sigma-Aldrich) or the uncoupler carbonyl cyanide p-trifluoromethoxyphenylhydrazone (CCCP, 10 µM; Sigma-Aldrich) was added. TMRM fluorescence analysis over the mitochondrial regions of interest was performed using ImageJ. A reduction in TMRM fluorescence represents mitochondrial membrane depolarization. In the graph bars, we indicated TMRM fluorescence after 30-min oligomycin administration in the cells of the indicated genotypes. The cells were always loaded in the presence of the multidrug resistance inhibitor cyclosporine H.

### Mitochondria morphology analysis

Quantification of mitochondria length was performed by using ImageJ software. To measure mitochondrial length, we created maximum-intensity projections of z-series with 0.2-µm increments. Quantification was then performed by using "Squassh" (Segmentation and QUAntification of Subcellular SHapes), a plugin compatible with the image processing software ImageJ or Fiji, freely available from http://mosaic.mpi-cbg.de/?q=downloads/imageJ. Squassh is a segmentation method that enables both colocalization and shape analyses of subcellular structures in fluorescence microscopy images (Rizk et al, 2014). For our analysis, segmentation was performed with the minimum intensity threshold set to 0.15 and the regularization weight to 0.015.

The mitochondria morphology score was assigned as in Pogson et al (2014). Briefly, a morpho score is assigned to each imaged cell according to the morphology of its mitochondrial network. Numbers represent the designated "morphology score": 0 = cell with a full complement of mitochondria; 1 = cell with a full complement of mitochondria and some clumped mitochondria; 2 = cell with a reduced mitochondrial network and some clumped mitochondria; 3 = cell with a clumped mitochondrial network; and 4 = cell with a complete clumped mitochondrial network.

### Total RNA extraction and qRT-PCR

Total RNA was extracted from *Drosophila* S2R+ cells using TRI Reagent (Sigma-Aldrich) according to the manufacturer's instructions. The RNA pellet was dissolved in 5–10 µl RNAase-free water. Total RNA was extracted from approximately 10 flies using Trizol (Life Technologies) and further purified by precipitation with LiCl 8M. RNA samples were checked for integrity by capillary electrophoresis (RNA 6000Nano LabChip; Agilent Technologies). For each sample, 1 µg of RNA was used for first-strand cDNA synthesis, using 10 µM deoxynucleotides, 10 µM oligo-dT, and SuperScript II (Life Technologies). qRT-PCRs were performed in triplicate in a 7500 Real-Time PCR System (Life Technologies) using SYBR Green

chemistry (Promega). The $2^{-\Delta\Delta Ct}$ (RQ, relative quantification) method implemented in the 7500 Real-Time PCR System software was used to calculate the relative expression ratio (ref.). The *USP8* oligonucleotides primer used were *USP8*_F (CACCCATTCAAATTGTCGAG) and *USP8*_R (TCGATGGTCTCAATGTCGTT). *Rp49* was used as endogenous control and the oligonucleotides used were *Rp49* F (ATCGGTTACG-GATCGAACAA) and R (GACAATCTCCTTGCGCTTCT).

### Drosophila *stocks and procedures*

*Drosophila* were raised under standard conditions at 25°C unless differently stated on agar, cornmeal, yeast food. *park*[25] mutants and UAS-Parkin have been described before (Greene et al, 2003). *PINK1*[B9] mutants (Park et al, 2006) were provided by Dr. J Chung (KAIST). *w*[1118] and *Act-GAL4* strains were obtained from the Bloomington *Drosophila* Stock Center. UAS-USP8 RNAi and UAS-Marf RNAi lines were obtained from the VDRC Stock Center. *Usp8*[−/+] and UAS-Usp8 (uspy) lines were kindly provided by S Goto (Mukai et al, 2010).

### *Climbing assays*

Climbing assays were performed as previously described (Greene et al, 2003). For the climbing assay upon drug treatment, groups of 10 flies were collected and placed into an empty vial (12 × 5 cm) with a line drawn at 6 cm from the bottom of the tube. The flies were gently tapped to the bottom of the tube, and the number of flies that successfully climbed above the 6-cm mark after 10 s was noted. Fifteen separate and consecutive trials were performed for each experiment, and the results were averaged. At least 40 flies were tested for each genotype or condition. Data collection and analysis were performed blind to the conditions of the experiments unless otherwise indicated.

### *Isolation of mitochondria*

Mitochondria were extracted from whole flies by differential centrifugation. Each sample was homogenized using a Dounce glass–glass potter and a loose-fitting pestle in a mannitol–sucrose buffer (225 mM mannitol, 75 mM sucrose, 5 mM Hepes, and 0.1 mM EGTA, pH 7.4) supplemented with 2% BSA. The samples were then centrifuged at 1,500 *g* at 4°C for 6 min. The pellet was discarded by filtering the sample through a fine mesh, and the supernatant was centrifuged at 7,000 *g* at 4°C for 6 min. The resulting pellet was resuspended in mannitol–sucrose buffer without BSA before being centrifuged at 7,000 *g* under the same conditions as above and resuspended in a small volume of mannitol–sucrose buffer. Protein concentration was measured using the biuret test.

### *Mitochondrial respiration*

Rates of mitochondrial respiration were measured using the Oxytherm System (Hansatech) with magnetic stirring and thermostatic control maintained at 25°C. Isolated *Drosophila* mitochondria (1 mg/ml) were incubated in 120 mM KCl, 5 mM P$_i$-Tris, 3 mM Hepes, 1 mM EGTA, and 1 mM MgCl$_2$, pH 7.2, and additions were made as indicated in the figure legends. O$_2$ consumption was calculated according to the slope of the registered graph and plotted as ng atoms: O$_2$ × min$^{-1}$ × mg$^{-1}$. RCR (ADP-stimulated respiration over basal respiration) was calculated.

### *Immunostaining of whole-mounted brains*

Brains of 15-d-old male control or mutant flies were dissected in ice-cold PBS and fixed in 4% PFA at room temperature for 20 min. Samples were washed six times for 10 min with PBS + 0.3% Triton X-100, permeabilized with PBS + 1% Triton X-100 for 10 min, and blocked with PBS + 0.3% Triton X-100 containing 1% BSA overnight at 4°C. For immunostaining of DA neurons, rabbit anti-TH antibody (Millipore) diluted 1:100 in PBS + 0.3% Triton X-100 containing 0.3% BSA was added and incubated over three nights at 4°C. Brains were washed and blocked again as described above, despite the blocking this time being carried out at RT for 1 h. The immunoreaction was revealed with Cy3-conjugated anti-rabbit IgG (Jackson Immuno-Research) at a working dilution of 1:500 in PBS + 0.3% Triton X-100 containing 0.3% BSA overnight at 4°C. After another six washing steps, whole brains were mounted with Vectashield (Vector Laboratories). Z-stack images were obtained by a Zeiss LSM700 confocal microscope.

### *Drug treatment*

The specific USP8 inhibitor DUBs-IN-2 (ChemScene LLC) was administered to flies in the food. DUBs-IN-2 (or DMSO) was diluted in water to the desired concentration and used to reconstitute dry Formula 4-24 Instant *Drosophila* Medium (Carolina Biological Supply). 1-d-old male mutant or control flies in groups of 10 were fed on the supplemented food for 48 h and subsequently climbing assay was performed. In the case of DA neuron staining and measurement of dopamine levels, mutant and control flies were aged for 15 d on the supplemented food that was exchanged every 2 d adding fresh drug or vehicle. The use of non-harmful food coloring demonstrated food uptake and excluded the possibility that smell or taste of the drug prevented the latter. Toxic concentrations were excluded beforehand by performing dose-dependent viability curves on control flies.

### Drosophila *head dopamine amount measurement (HPLC)*

*Drosophila* heads of 15-d-old male flies were dissected out and collected separately in 10 µl of ice-cold 0.2 N perchloric acid. The tissue was homogenized by sonication for 15 s and kept on ice for 20 min, then centrifuged at 12,000 *g* for 10 min, and the supernatant was collected. The samples were further diluted and 5 µl was injected into a HPLC system equipped with a rheodyne injector and a guard cell, set to +350 mV (E$_1$ = +150 mV, E$_2$ = −350 mV, s: 2 nA). A C$_{18}$ ion-pair, reverse phase analytical column (4.6 × 250 mm; 5 µm particle size; Agilent Technologies) was used for the separation of biogenic amines with a flow rate of 0.8 ml/min. Composition of the mobile phase was 75 mM sodium phosphate monobasic monohydrate, 6% acetonitrile, 1.7 mM 1-octane sulfonic acid, and 25 µM EDTA (pH 3 ± 0.01). Dopamine values were determined by comparing with the standard peak value.

### *Electron microscopy*

Thoraces were prepared from 3-d-old adult flies and fixed overnight in 2% paraformaldehyde and 2.5% gluteraldehyde. After rinsing in 0.1 M cacodylate buffer with 1% tannic acid, the samples were postfixed in 1:1 2% OsO$_4$ and 0.2 M cacodylate buffer for 1 h. The samples were rinsed, dehydrated in an ethanol series, and

embedded using Epon. Ultrathin sections were examined using a transmission electron microscope.

### Life span analysis

Male flies of the indicated genotypes were collected during 12 h after hatching and grouped into 20 flies per food vial. At least 100 flies were used for the analysis (exact numbers are indicated in the figure legends). The flies were transferred to fresh food (and fresh drug for the inhibitor treatment) every 2 d, and dead flies were counted in the same interval.

### Measurement of food uptake

Dry Formula 4-24 Instant *Drosophila* Medium (Carolina Biological Supply) was reconstituted with a mix of water and food-coloring patent blue V (E131) (1:1) previously supplemented with DMSO or the desired DUBs-IN-2 concentration. Three groups of 10 male 1–3-d-old *w1118* flies were kept in the food vials for 48 h. Afterward, the flies were weighed and homogenized in 20 volumes of PBS with an electric potter. The homogenate was centrifuged for 10 min at 15.000 *g* and absorbance of the supernatant was measured at 640 nm.

### BN PAGE

Pellets of mitochondria isolated from adult male flies of the indicated genotypes were suspended at 10 mg × ml$^{-1}$ in 1× native PAGE sample buffer (Invitrogen) supplemented with protease inhibitor mixture (Sigma-Aldrich), solubilized with 2% (wt/vol) digitonin and immediately centrifuged at 100,000 *g* for 25 min at 4°C. The supernatants were supplemented with native PAGE 5% G-250 sample additive (Invitrogen) and quickly loaded onto a blue native polyacrylamide 3–12% gradient gel (Invitrogen). After electrophoresis, the gels were fixed in 50% methanol + 10% acetic acid for 20 min at RT, stained in 0,025% Coomassie + 10% acetic acid overnight at RT and destained with 10% acetic acid.

### Sperm content and reproductive apparatus viability assay

The anatomy of male reproductive apparatus was analyzed on 10 males per group. To this aim, the reproductive apparatus was removed, placed on a slide with few drops of *Drosophila* Ringer's solution (182 mM KCl, 46 mM NaCl, 3 mM CaCl$_2$ 2H$_2$O, and 10 mM Tris–HCl, pH 7.2) and freshly examined under a light microscope. To verify the presence of sperm inside the seminal vesicles, these were then removed from the whole apparatus and gently punctured with a needle to let the sperm pouring out. Five more intact apparatuses per group were stained with a dead/alive cell viability kit (Molecular Probes) that allows differentiation between live green cells, permeable to green SYBR 14 nucleic acid stain, and red dead cells, permeable to propidium iodide nucleic acid stain, which penetrates through compromised membranes.

### Statistical analysis

Where multiple groups were compared, statistical significance was calculated by one-way or two-way ANOVA with a post hoc Tukey or Dunett correction. All statistical significance was calculated at *P* = 0.05, using GraphPad Prism 8. For all the analysis, the samples were collected and processed simultaneously and, therefore, no randomization was appropriate (GraphPad Prism. ****$P$ < 0.0001, ***$P$ < 0.001, **$P$ < 0.01, and *$P$ < 0.05). Please refer to the enclosed document for detailed statistical tests.

## Supplementary Information

## Acknowledgements

This work was supported by grants from the Italian Ministry of Health "Ricerca Finalizzata" (GR-2011-02351151), Rita Levi Montalcini "Brain Gain" program, and Michael J Fox RRIA 2014 (Grant ID 9795) to E Ziviani and by ERC FP7-282280, FP7 CIG PCIG13-GA-2013-618697, and Italian Ministry of Research FIRB RBAP11Z3YA_005 to L Scorrano. AJ Whitworth is funded by MRC Core funding (MC_UU_00015/6). We would like to acknowledge Francesco Boldrin from the EM facility for the help and technical support. We thank the Sheffield RNAi Screening Facility, Biomedical Sciences, University of Sheffield, for providing the RNAi library and reagents used in this study supported by the Wellcome Trust (grant reference number 084757)"

### Author Contributions

S von Stockum: data curation, formal analysis, validation, investigation, methodology, and writing—review and editing.
A Sanchez-martinez: data curation, formal analysis, validation, investigation, and methodology.
S Corrà: data curation, formal analysis, methodology, and writing—review and editing.
J Chakraborty: formal analysis and methodology.
E Marchesan: data curation and methodology.
L Locatello: conceptualization, formal analysis, and methodology.
C Da Rè: conceptualization, data curation, formal analysis, and methodology.
P Cusumano: conceptualization and methodology.
F Caicci: data curation and methodology.
V Ferrari: methodology.
R Costa: supervision and writing—review and editing.
L Bubacco: conceptualization, data curation, formal analysis, and writing—review and editing.
MB Rasotto: conceptualization, data curation, formal analysis, supervision, and methodology.
I Szabo: conceptualization, supervision, and writing—review and editing.
AJ Whitworth: conceptualization, data curation, formal analysis, supervision, funding acquisition, methodology, and writing—review and editing.
L Scorrano: conceptualization, data curation, formal analysis, funding acquisition, methodology, and writing—review and editing.
E Ziviani: conceptualization, data curation, formal analysis, supervision, funding acquisition, methodology, writing—original draft, and project administration.

## Conflict of Interest Statement

The authors declare that they have no conflict of interest.

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
