## [Reviewer comments · Life Science Alliance]

Life Science Alliance

Inhibition of the deubiquitinase USP8 corrects a *Drosophila* PINK1 model of mitochondria dysfunction

Elena Ziviani, Sophia vonStockum, Alvaro Sanchez-Martinez, Samantha Corrà, Joy Chakraborty, Elena Marchesan, Lisa Locatello, Caterina Da Rè, Paola Cusumano, Federico Caicci, Vanni Ferrari, Rodolfo Costa, Luigi Bubacco, Maria Berica Rasotto, Ildiko Szabo, Alexander Whitworth, and Luca Scorrano

DOI: <https://doi.org/10.26508/lsa.201900392>

Corresponding author(s): Elena Ziviani, Department of Biology, University of Padova, Padova, Italy

Review Timeline:

Submission Date:	2019-04-01
Editorial Decision:	2019-04-02
Revision Received:	2019-04-04
Accepted:	2019-04-05

Scientific Editor: Andrea Leibfried

Transaction Report:

Please note that the manuscript was previously reviewed at another journal and the reports were taken into account in the decision-making process at *Life Science Alliance*. Since the original reviews are not subject to *Life Science Alliance*'s transparent review process policy, the reports and author response cannot be published.

April 2, 2019

RE: Life Science Alliance Manuscript #LSA-2019-00392-T

Dr. Elena Ziviani
Department of Biology, University of Padova
via ugo bassi 58/B
Padova 35121
Italy

Dear Dr. Ziviani,

Thank you for submitting your revised manuscript entitled "Inhibition of the deubiquitinase USP8 corrects a Drosophila PINK1 model of mitochondria dysfunction" to Life Science Alliance. Your work was previously reviewed at another journal, and you provided us with the reviewer reports obtained as well as your response to the remaining concerns.

As outlined to you prior to submission to our journal, we appreciate your work and agree with the reviewers that your findings are interesting. The reviewers noted that protein abundance analysis, quantifications, statistics as well as the analyses in the various mutant backgrounds would need to get further revised/extended, and that it remains unclear whether USP8 has direct or indirect effects on MFN / Marf levels. They also requested further validation of the USP8 RNAi approach and an analysis of the specificity of the inhibitor used. I think that you have responded to all of these concerns in a good way and appreciate the introduced changes. I would thus be happy to publish your paper in Life Science Alliance pending final revisions necessary to meet our formatting guidelines:

- please note the missing axes for the oxygen consumption analyses in Fig 4
- please note inconsistency in labeling in Fig S5 (Marf vs MFN)
- please add scale bars to Fig 1A, D, E, 2C
- please list 10 authors et al in the reference list
- please list all contributing authors in our submission system

A. FINAL FILES:

B. MANUSCRIPT ORGANIZATION AND FORMATTING:

Sincerely,

Andrea Leibfried, PhD
Executive Editor

Life Science Alliance
Meyerohofstr. 1
69117 Heidelberg, Germany
t +49 6221 8891 502
e a.leibfried@life-science-alliance.org
www.life-science-alliance.org

April 5, 2019

RE: Life Science Alliance Manuscript #LSA-2019-00392-TR

Dr. Elena Ziviani
Department of Biology, University of Padova, Padova, Italy
via ugo bassi 58/B
Padova 35121
Italy

Dear Dr. Ziviani,

Thank you for submitting your Research Article entitled "Inhibition of the deubiquitinase USP8 corrects a Drosophila PINK1 model of mitochondria dysfunction". It is a pleasure to let you know that your manuscript is now accepted for publication in Life Science Alliance. Congratulations on this interesting work.

DISTRIBUTION OF MATERIALS:

Again, congratulations on a very nice paper. I hope you found the review process to be constructive and are pleased with how the manuscript was handled editorially. We look forward to future exciting submissions from your lab.

Sincerely,
